

# A comprehensive analysis and performance evaluation for osteoporosis prediction models

Zahraa Noor Aldeen M. Shams Alden[1,2,*] and Oguz Ata[3,*]

[1] Faculty of Tourism Science, University of Kerbala, Kerbala, Iraq
[2] Department of Electrical and Computer Engineering, Altinbas University, Istanbul, Turkey
[3] Department of Software Engineering, Engineering and Architecture Faculty, Altinbas University, İstanbul, Turkey
[*] These authors contributed equally to this work.

Corresponding author
Zahraa Noor Aldeen M. Shams Alden,
zahraa.noor@uokerbala.edu.iq,
203720159@ogr.altinbas.edu.tr

## ABSTRACT

Medical data analysis is an expanding area of study that holds the promise of transforming the healthcare landscape. The use of available data by researchers gives guidelines to improve health practitioners' decision-making capacity, thus enhancing patients' lives. The study looks at using deep learning techniques to predict the onset of osteoporosis from the NHANES 2017–2020 dataset that was preprocessed and arranged into SpineOsteo and FemurOsteo datasets. Two feature selection methods, namely mutual information (MI) and recursive feature elimination (RFE), were applied to sequential deep neural network models, convolutional neural network models, and recurrent neural network models. It can be concluded from the models that the mutual information method achieved higher accuracy than recursive feature elimination, and the MI feature selection CNN model showed better performance by showing 99.15% accuracy for the SpineOsteo dataset and 99.94% classification accuracy for the FemurOsteo dataset. Key findings of this study include family medical history, cases of fractures in patients and parental hip fractures, and regular use of medications like prednisone or cortisone. The research underscores the potential for deep learning in medical data processing, which eventually opens the way for enhanced models for diagnosis and prognosis based on non-image medical data. The implications of the study shall then be important for healthcare providers to be more informed in their decision-making processes for patients' outcomes.

## INTRODUCTION

Osteoporosis is a health issue where bones weaken and become fragile, making fractures more likely to happen. It develops gradually over time and is often only detected when a bone breaks due to impact or injury such as a fall. For a very long period, osteoporosis (OP) was only associated with a decrease in the total density of bones, which decreased bone mass and increased fracture risk (*Mornar et al., 2024*). The structure inside a bone

can be likened to a honeycomb. In individuals with osteoporosis, the bone "walls" of this honeycomb structure shrink while the spaces between the bones widen. Moreover, the outer layer of the bone thins out, leading to weakened bone (*Yang et al., 2013*).

The most common medical image acquisition are computed tomography (CT), magnetic resonance imaging (MRI), and dual X-ray DXA which have been used in medical prediction models (*Sistaninejhad, Rasi & Nayeri, 2023*; *Lakshmipriya, Pottakkat & Ramkumar, 2023*), but dual X-ray absorptiometry was regarded as the key standard method for determining the patient's bone mineral density (*Molino et al., 2020*) as it is affordable, less radiation to the patients, and is widely available in most medical centers. This article aims to analyze medical data collected from the NHANES 2017–March 2020 (*NHANES, 2020*) which includes data collected from DXA images of the spine and femur in addition to the dataset acquired from questionnaire reports of patients who may or may not suffer from osteoporosis.

Deep neural networks are used in medical deep learning, a subset of artificial intelligence (AI), to analyze medical data and produce predictions or diagnoses. In the field of healthcare, deep learning distinguishes itself from rule-based or expert systems by making decisions based on identifying patterns that may go unnoticed by humans (*Nasir et al., 2023*). These algorithms are particularly skilled at detecting indications of diseases or irregularities, proving valuable in tasks such as analyzing images. Their ability to adapt to data and continuously improve makes them well-suited for applications offering accuracy and cost reduction. By supporting treatment plans, deep learning has the potential to transform healthcare through the development of predictive models (*Merdas & Mousa, 2023*).

According to an analysis of deep learning algorithms in 2022, there has been a noticeable increase in the number of review and survey papers published on this topic over the past three to four years (*Egger et al., 2022*).

Over this period, there has been a release of reviews on medical deep learning typically around once a month. The analysis also found that the use of deep learning algorithms in clinical settings has made a significant impact on the field of medicine (*Mousa et al., 2020*). This detailed analysis delves into the progress made in deep learning technologies and underscores their capacity to enhance outcomes.

The study's main contributions are presented as follow:

1. This study demonstrates the potential application of deep learning algorithms in medical data processing and improving diagnostic and prognostic models based on non-image medical data. The output would be helpful for medical experts in making decisions that are informed, hence bettering the health outcomes of patients.

2. A related work is provided to illustrate an overview of related research using deep learning techniques to analyze medical data.

3. This study highlights the importance of feature selection methods to best improve the accuracy of the prediction models. These models illustrate how they may be applied in medical practice and reduce the risk of any bone fracture.

4. The article highlights family medical history as a vital aspect in predicting osteoporosis by paying close attention to such cases as fractures in the patient and hip fractures in the parents.

5. A comparison of our developed model to another model developed by researchers in *Alalhareth & Hong (2023)* to indicate the importance of the feature selection methods and deep learning techniques.

## RELATED WORK

### Current status

Many researchers have attempted using machine learning techniques to predict osteoporosis and showed promising results in this field.

A study conducted by *Kranthi, Sailaja & Jyothi (2024)* that uses deep learning methods, particularly convolutional neural networks (CNNs), to automate the interpretation and analysis of images by leveraging extensive medical image datasets. Their process includes converting raw image data into structured formats to facilitate accurate analysis. Their proposed model achieved an F1-score of 93.22% and a classification accuracy of 91.16%. However, the authors stated some limitations such as overfitting and the complexity of training the model. These issues highlight the need for optimization and substantial computational resources to enhance model performance and reliability.

*Sisodia, Nayak & Boghey (2024)* developed a hybrid CNN-deep neural network (DNN) model to predict stock and index prices for Bank Nifty, leveraging historical trading data for feature extraction and accurate forecasting and achieving a prediction accuracy of 97.48%. Despite the high accuracy, the authors highlight some of the limitations such as overfitting, the need for robust computational resources, and the complexity of training a hybrid model.

Another recent study conducted in 2021 highlighted the effectiveness of a learning network known as DeepDXA in evaluating bone mineral density (BMD) using pelvis X-rays (*Ho et al., 2021*). By examining data within the X-ray images, this model can predict BMD values with precision. DeepDXA presents an alternative in comparison to conventional methods like dual-energy X-ray absorptiometry (DXA), which necessitates specialized equipment and trained personnel. Moreover, DeepDXA has demonstrated performance in predicting osteoporosis from hip X-ray images with an accuracy of 88%, allowing for cost-effective identification of osteopenia and osteoporosis (*Ho et al., 2021*). The findings of this research could have an impact on the accuracy and ease of diagnosing osteoporosis, potentially improving outcomes by enhancing the precision and usability of osteoporosis diagnosis in clinical environments. Other related work can be summarized in Table 1. The purpose of this comparison to related work is to emphasize the advancements in machine learning and deep learning techniques applied to osteoporosis prediction and similar medical data analysis tasks. By doing so, the study aims to highlight the existing gaps and limitations in current methods that it seeks to address, showcasing the effectiveness and novelty of our approach.

**Table 1 Related work.**

| No. | Authors | Year | Contribution |
|---|---|---|---|
| 1. | Y. Singh, V. Atulkar, J. Ren, J. Yang, H. Fan, L. Latecki and H. Ling | 2021 | Deep learning was utilized in this study to predict bone mineral density from dental panoramic radiographs, which has the potential to be used as a tool for osteoporosis screening. This research presents the evaluation of a DCNN-based CAD system for the detection of osteoporosis using dental panoramic radiographs, achieving 87.86% accuracy with the inclusion of age data for BMD prediction (*Singh et al., 2021*). |
| 2. | I. Wani and S. Arora | 2020 | Examined deep Learning techniques alongside energy X-ray absorptiometry (DXA) to predict osteoporotic. This was a review article that studies the implementation of neural network models to assist in Osteoporosis prediction (*Wani & Arora, 2020*). |
| 3. | Giulia Molino, Giorgia Montalbano, Carlotta Pontremoli, Sonia Fiorilli ,and Chiara Vitale Brovarone | 2020 | Outline the advantages of using complex imaging techniques such as MicoCT, which provide high resolution for Osteoporosis diagnosis. **Limitation**: the authors state the limitation of their study as these techniques produce higher doses of radiation and have higher costs, making them difficult to use in medical centers (*Molino et al., 2020*). |
| 4. | G. Long, C. Liu, T. Liang, Z. Zhang, Z. Qin and X. Zhan | 2023 | A systematic review providing an assessment and meta-analysis of the diverse machine learning methods employed in predicting osteoporotic fractures. This meta-analysis identified, from ten studies involving over 1.2 million participants, significant predictors of fracture risk in postmenopausal women, including age, BMI, reproductive history, and use of vitamin D. The results support a focus on high-risk individuals for prevention strategies (*Long et al., 2023*). |
| 5. | Efat Jabarpour, Amin Abedini, and Abbasali Keshtkar | 2020 | This study used data mining methods to predict the risk of Osteoporosis. Their findings could be helpful to be used as a sample for future Osteoporosis prediction in new patients. **Limitation:** the study used data from 2006–2010 which affects the implication of their finding on the current population. Also, the models' accuracy was lower than any other models and needs more improvement to increase their accuracy (*Jabarpour, Abedini & Keshtkar, 2020*). |
| 6. | J. Smets, E. Shevroja, T. Hügle, W. Leslie and D. Hans | 2021 | Researchers investigate machine learning algorithms like decision trees, support vector machines, and artificial neural networks for predicting osteoporosis. Among 89 studies, supervised learning models coupled with medical imaging in risk prediction were highlighted, though data quality and external validation challenges remain. Recommendations pertain to the use of standardized checklists to improve reproducibility (*Smets et al., 2021*). |

**Table 1** (*continued*)

| No. | Authors | Year | Contribution |
|---|---|---|---|
| 7. | Kopperdahl, David L., Thor Aspelund, Paul F. Hoffmann, Sigurdur Sigurdsson, Kristin Siggeirsdottir, Tamara B. Harris, Vilmundur Gudnason, and Tony M. Keaveny. | 2019 | Machine learning methods were employed to forecast risks, for women with osteoporosis. The investigation found that machine learning systems showed accuracy levels to those of methods used to evaluate fracture risks (*Kopperdahl et al., 2014*). **Limitation**: DXA was not used in this study; instead, a CT-based measure of areal BMD was used, which may not perfectly correlate with DXA measurements. Also, this design limits the ability to estimate prevalence and absolute risk directly. |
| 8. | M. A. Alsheikh, A. Selamat, and M. A. Al-Masni | 2019 | Investigate the application of data mining techniques in diagnosing osteoporosis, including clustering, classification, and association rule mining (*Alsheikh, Selamat & Al-Masni, 2019*). |

## Limitations

Recent studies have proven the significant advancement in osteoporosis prediction made by machine learning based on image and non-image medical data and questionnaires of patients. Fusing a wide range of clinical records, genetic data, and patient-reported outcomes has increased the predictive precision of ML models.

1. **Data integration and standardization**: Combining data integration from a mix of sources such as clinical records, genetic data, and questionnaires into one harmonious format remains a significant challenge due to discrepancies in data formats and standards. This requires effective techniques in data fusion (*Ebbehoj et al., 2022*).

2. **Privacy and security**: Medical data are highly sensitive and cannot be exposed to the public domain. Among these, data handling and processing for protection must comply with regulations such as HIPAA and GDPR (*Vitabile et al., 2019*).

3. **Model interpretability**: ML models, especially deep learning models, are considered a black box. The interpretability of such models and their predictions is quite crucial for clinical acceptance and trust (*Cui et al., 2023*).

4. **Generalizability and bias**: ML models being trained on specific datasets risk generalizing on other populations due to differences in demographic characteristics and clinical practices. Correcting biases in training data is essential, hence these models must be validated on diverse datasets (*Amal et al., 2022*).

Our study effectively addresses several key limitations identified in the current research for using machine learning (ML) techniques to predict osteoporosis based on non-image medical data and patient questionnaires.

1. **Data integration and standardization**: Several datasets of femur, spine, and patient questionnaire data obtained from the NHANES 2017–2020 were integrated into our research. By merging the different datasets and incorporating feature selection techniques such as recursive feature elimination (RFE) and mutual information (MI), this study makes sure that the data used is complete and relevant to our research. This approach is mainly instrumental in mitigating the issues that result from the heterogeneity and standardization of data within multi-modal data integration.

2. **Privacy and security**: The fact that NHANES data is an openly accessed dataset would mean that its uses should be subject to all data protection regulations. Future work may be strengthened by explicitly addressing how privacy and security are maintained, particularly in line with guidelines set by HIPAA and GDPR, to enhance trust and applicability for real-world clinical settings.

3. **Interpretability of models**: Our research makes use of sequential deep neural networks (DNN), convolutional neural networks (CNN), and recurrent neural networks (RNN) for prediction. The authors provide transparency in how such models work and which features are most influential by detailing the architecture and process of feature selection. This type of transparency improves the interpretability of models, which is a critical limitation in the deployment of ML in clinical practice.

4. **Generalizability and bias**: Using a large, diverse dataset from NHANES helps to enhance the generalizability of our findings. However, our study does point out the limitation of working with data from a particular demographic (the Hispanic population) which could not capture all variations in patient demographics. Future research may do this by targeting more diverse populations to ensure the robustness of the model in a different demographic group.

## METHODOLOGY

### Data preparation

One of the first data preparation steps is to ensure there is no missing or incorrect data that could affect the decision-making process. The dataset size affects the process of processing and analyzing (*Kopperdahl et al., 2019*). The datasets used in this article were NHANES 2017–2020 (*NHANES, 2020*), and are of the Hispanic population who are more likely to have osteoporosis than any other ethnicity, making them a great selection for this study. The Femur, Spine, and Questionnaire datasets were collected from the CDC Center for Disease Control and Prevention. After first analyzing the data and comparing these three datasets, some data rows were discarded since some of those patients were not included in the questionnaire data. This could be because those patients refused to participate in the questionnaire, or their information was incorrectly collected. The Femur dataset contains 3,545 rows, while the Spine dataset contains 1,889 data rows.

After comparing these datasets to the questionnaire dataset, a few data rows were omitted from the resulting dataset as not all patients participated in the survey that was conducted by CDC center, National Center for Health Statistics (*CDC, 2021b*). Only 1,725 and 3,544 data rows for spine and femur datasets, respectively, remain in this study. The resulting datasets were only two datasets which were a result of the merging process of the Spine and Femur datasets with the Questionnaire dataset based on a common identifier, which is the patients' sequence number. These two resulting datasets were SpineOsteo and FemurOsteo Datasets.

All NHANES 2017–2020 sample individuals 50 years of age or older were eligible. Pregnant women were not allowed to take the DXA test. Participants who were not pregnant but had been omitted from the test were eligible non-respondents. The following are the reasons for being disqualified from the DXA test:

- Pregnancy (confirmed by a pregnancy test).
- Self-reported prior week exposure to radiographic contrast chemicals such as barium or dyes.
- Weight exceeding 450 pounds that was measured (DXA table restriction).

The Shepherd Lab assessed and evaluated each participant and phantom scan using conventional radiologic methods and study-specific protocols created for the NHANES (*NHANES, 2023*). All femur scans performed between 2017 and March 2020 were analyzed using the Hologic software, APEX v4.0 (Hologic). The correctness and consistency of the findings were confirmed by the Shepherd Research Lab's expert examination of 100% of the participant scans that were subjected to analysis (*NHANES, 2023*).

### Femur dataset

In this section, a discussion about the type of data in this dataset will be presented. If a participant had both hips broken, had both hips replaced, or had pins in both hips, they were disqualified from the femur scan.

When assessing for osteoporosis, particular attention is given to the hip unless there has been a fracture, hip replacement, or surgery involving the left hip. In some cases, scans are conducted on the other hip to gather information on bone mineral density (BMD), bone mineral content (BMC), bone area measurements, and other relevant data. These data were extracted from DXA imaging of the selected area (*NCHS, 2020*). Only 1,048 missing values of data were not recorded.

### Spine dataset

The Spine data set includes bone measurements for the total spine and vertebrae L1–L4 (*CDC, 2021b*), including:

- Bone mineral content (BMC).
- Bone area measurements.
- Bone mineral density (BMD).

The missing value for most of these data was 1,816 value, except for calculated vitamin K and calculated vitamin D where the number of missing values was 690.

### Questionnaire dataset

This dataset includes answered questions from the patients that were included in this data collection process (*CDC, 2021a*). These questionnaires were summarized as follows:

- Information on a spine, wrist, or hip fracture that was self-reported.
- Age at each spine, wrist, or hip fracture and the total number of fractures
- The severity of each fracture of the hip, wrist, or spine
- Other bone fractures, the severity of the trauma that caused them, and the age at which they happened.
- Whether the individual has ever received an osteoporosis diagnosis (this data was used as target variable since it specifies if the patient had or had not have Osteoporosis).

- Whether the participant has had treatment for osteoporosis using a prescription drug, such as Actonel, Boniva, Fosamax, or Forteo.
- Whether or not the participant's biological parents had osteoporosis diagnoses.
- Whether the participant's parents ever suffered a hip fracture, and if so, when.

## Feature selection

Selecting the best features that have a high correlation with the target variable is a very important step in the process of analyzing and building a classification model (_Chen et al., 2019_). There are many supervised and unsupervised algorithms for feature selection, as shown in Fig. 1, where the supervised approach uses the target label or variable, unlike the unsupervised algorithm, which doesn't need access to the target variable (_Oleszak, 2023_).

- Recursive feature elimination (RFE) selects features in a recursive manner by training the model and eliminating the least significant feature(s) at each iteration (_Samb et al., 2012_). It assesses the importance of characteristics in relation to the model's performance. The least significant characteristics were eliminated, and the classification features were repeatedly updated (_Chen et al., 2018_). It is a wrapper feature selection algorithm that uses a filter-based feature selection algorithm internally to build an efficient classifier. In this article, RFE is based on the random forest algorithm RF-RFE, which was presented by _Breiman (2001)_ and was designed to select only 30 variables of the datasets for all models.
- Mutual information (MI) serves as a method for uncovering the connections between variables (_Priscilla & Prabha, 2021_). It explains how to determine the significance of a group of features with the target variable. When the MI value is greater than zero for two variables, they are considered statistically related; if the MI value is less than zero, they are deemed statistically independent. The MI is directly linked to the entropies of the variables (_Estévez et al., 2009_). In these models, only the 30 features with correlations were selected using this algorithm.

## Deep learning models

Deep learning is a subset of artificial intelligence and machine learning. Recent years have seen major advances in this subfield, leading to the development of tools for the comprehensive analysis of complicated medical data. Applying deep learning to the medical sector began in the early 2010s (_Litjens et al., 2017_), with the highly successful application of DL to image recognition. Significant performances from CNNs have been reported for medical image analysis, allowing automatic detection and classification of diseases from medical imaging data like X-rays, MRIs, and CT scans (_McKinney et al., 2020_). The use of machine learning algorithms in the health sector has increased over time which indicates the powerfulness of these algorithms in analyzing medical data and feature selection techniques to assist in diagnosis and disease prediction models (_Rana & Bhushan, 2023_; _Salehi et al., 2023_). The importance of deep learning methods in various medical image analysis tasks such as segmentation and reconstruction has significantly improved over time by providing higher accuracy and adaptability compared to traditional methods (_Gong et al., 2024_). The models used are Sequential DNNs, CNNs, and RNNs.

**Feature Selection Methods**

**Supervised**
- **Embedded**
  - LASSO
  - Auto-encoder with bottleneck
- **Filters**
  - Pearson's r
  - Kendall Tau
  - Spearman's Rho
  - Chi2
  - Mutual Info
  - F-score
  - Point-biserial
- **Wrappers**
  - Forward selection
  - Forward selection
  - Recursive Feature Elimination

**Unsupervised**
- Drop incomplete features
- Drop features with (near-)zero variance
- Drop features with high multicollinearity

**Figure 1** **Feature selection methods.**

Each of these models has a unique set of strengths and applications, especially regarding medical diagnostics.

1. **Sequential deep neural networks (DNN):** This is a kind of neural network in which neurons are applied layer by layer and each layer processes input data to extract features and make predictions. It has a better performance for structured data and has also been

used in much medical research works for predictive analytics (*Denoyer & Gallinari, 2014*).

2. **Convolutional neural networks (CNN):** Although majorly used in image analysis, they can still be applied for structured data. They use convolutional layers to detect local patterns, making them suitable for identifying intricate relationships within medical datasets (*Mishra, 2020*).

3. **Recurrent neural networks (RNN):** RNNs are created in a way to maintain a 'memory' of previous inputs to be able to handle sequential data. Long short-term memory (LSTM) units can capture long-term dependencies within data and are therefore extremely effective for time-series medical data (*Jayawardhana, 2020*).

This article discusses exploring the capabilities of learning methods such as DNN, CNN, and RNN in handling medical image data. Our study focuses on categorizing a target variable using a well-structured medical dataset. The constructed models will explore the selected variables from the dataset that are most correlated with the target variable, which in this article is the classification of patients with or without osteoporosis. The data were split into training, validation, and testing sets.

In this article, six models were designed to analyze these datasets using deep learning techniques that are most suitable for the analysis of medical data to develop classification models. These six models were Sequential DNN, CNN, and RNN, and each of these models used two different feature selection algorithms. Notably, all the models utilize the entropy loss function, employ the Adam optimizer, and measure performance using accuracy as the evaluation metric.

# EXPERIMENTAL WORK

## Extracted features

In this article, two feature selection algorithms were used to better understand which features have more connection with each other and would benefit the model by using the most suitable features out of 139 features in the datasets. These two algorithms were MI and RFE. The reasons for choosing these algorithms depend greatly on the type of our model and the datasets. RFE and MI methods are suitable for classification models and non-image datasets as they rank the importance of features based on the model's performance. In addition, MI and RFE algorithms are suitable for analyzing complex and nonlinear data, such as the datasets in this research, since they take into account the nonlinear correlations between features and class labels.

These two algorithms were used to extract important features to build the three classification models. The extracted features are shown in Tables 2 and 3 from SpineOsteo and FemurOsteo, respectively.

Using deep learning to extract important features from the two datasets combined shows how important the role of deep learning algorithms is in this process to ensure that only important features are selected to be trained in the DNN, CNN, and RNN models in this study. The analysis shows that most selected features were from the Femur and Spine datasets (64%) with the RFE algorithm and (54%) with the MI algorithm, and the remaining features were from the Questionnaire dataset.

**Table 2  Selected features from the SpineOsteo dataset.**

| No. | Recursive feature elimination (RFE) | | Mutual information classification (MI) | |
| --- | --- | --- | --- | --- |
| | Abbreviation | Feature's name | Abbreviation | Feature's name |
| 1. | DXXOSBMC | Total spine BMC | DXXOSBMC | Total spine BMC |
| 2. | DXXOSA | Total spine Area | DXXOSA | Total spine Area |
| 3. | DXXL1BMD | L1 BMD | DXXL1BMD | L1 BMD |
| 4. | DXXL1BMC | L1 BMC | DXXL1BMC | L1 BMC |
| 5. | DXXL1A | L1 Area | DXXL1A | L1 Area |
| 6. | DXXL2BMD | L2 BMD | DXXL2BMD | L2 BMD |
| 7. | DXXL2BMC | L2 BMC | DXXL2BMC | L2 BMC |
| 8. | DXXL2A | L2 Area | DXXL2A | L2 Area |
| 9. | DXXL3BMD | L3 BMD | DXXL3BMD | L3 BMD |
| 10. | DXXL3BMC | L3 BMC | DXXL3BMC | L3 BMC |
| 11. | DXXL3A | L3 Area | DXXL3A | L3 Area |
| 12. | DXXL4BMD | L4 BMD | DXXL4BMD | L4 BMD |
| 13. | DXXL4BMC | L4 BMC | DXXL4BMC | L4 BMC |
| 14. | DXXL4A | L4 Area | DXXL4A | L4 Area |
| 15. | OSQ040BA | First-time wrist fracture younger/older than 50? | OSQ040BA | First-time wrist fracture younger/older than 50? |
| 16. | OSD050BA | What reasons for a wrist fracture in 1st time? | OSD050BA | What reasons for a wrist fracture in 1st time? |
| 17. | OSQ130 | Ever taken prednisone or cortisone daily | OSQ130 | Ever taken prednisone or cortisone daily |
| 18. | OSQ160A | Mother had osteoporosis | OSQ160A | Mother had osteoporosis |
| 19. | OSQ160B | Father had osteoporosis | OSQ160B | Father had osteoporosis |
| 20. | DXXOSBMD | Total spine BMD | OSD030CA | Age when fractured spine 1st time |
| 21. | DXASPND0 | Calculated value of Vitamin D | OSQ040CA | Under/over 50 when fracd. spine 1st time |
| 22. | OSQ010A | Broken or fractured a hip | OSQ090F | Fracture result of severe trauma? |
| 23. | OSQ010C | Broken or fractured spine | OSQ040CB | Under/over 50 when fracd. spine 2nd time |
| 24. | OSQ020C | Times broken/fractured spine | OSD110C | Age when fracture occurred? |
| 25. | DXASPNK | Calculated value of Vitamin K | OSQ090E | Fracture result of severe trauma? |
| 26. | OSQ040BB | Under/over 50 when fracd. wrist 2nd time | OSD030CB | Age when fractured spine 2nd time |
| 27. | OSD110A | How old when fracture occurred? | OSQ090I | Fracture result of severe trauma? |
| 28. | OSQ120A | Any other fractures? | OSQ040AA | First-time hip fracture younger/older than 50? |
| 29. | OSQ140U | Duration of using prednisone or cortisone? | OSQ180 | Mother's age when fractured hip |
| 30. | OSQ150 | Have parents had osteoporosis? | OSQ210 | Father's age when fractured hip |

The existence of 19 similar selected features in both RFE and MI for the spine dataset and 21 similar selected features in both RFE and MI for the femur dataset indicates these features are more important or relevant in both datasets. These qualities are consistently identified as significant by both feature selection methods, indicating that they may give helpful information for the classification process. The overlap in selected features between the two algorithms implies that these features follow similar patterns and interact with the target variable across numerous feature selection procedures. This suggests that these features may have significant discriminating power and might be reliable markers in classification models for distinguishing between distinct groups. Figures 2 and 3 illustrate the selected features resulting from MI and RFE, respectively. As can be noticed from Fig. 2,

**Table 3  The selected features from the FemurOsteo dataset.**

| | Recursive feature elimination (RFE) | | Mutual information classification (MI) | |
|---|---|---|---|---|
| | **Abbreviation** | **Feature's name** | **Abbreviation** | **Feature's name** |
| 1. | DXXOFBMC | Total femur BMC | DXXOFBMC | Total femur BMC |
| 2. | DXXOFA | Total femur area | DXXOFA | Total femur area |
| 3. | DXXNKBMD | Femoral neck BMD | DXXNKBMD | Femoral neck BMD |
| 4. | DXXNKBMC | Femoral neck BMC | DXXNKBMC | Femoral neck BMC |
| 5. | DXXNKA | Femoral neck Area | DXXNKA | Femoral neck Area |
| 6. | DXXTRBMD | Trochanter BMD | DXXTRBMD | Trochanter BMD |
| 7. | DXXTRBMC | Trochanter BMC | DXXTRBMC | Trochanter BMC |
| 8. | DXXTRA | Trochanter Area | DXXTRA | Trochanter Area |
| 9. | DXXINBMD | Intertrochanter BMD | DXXINBMD | Intertrochanter BMD |
| 10. | DXXINBMC | Intertrochanter BMC | DXXINBMC | Intertrochanter BMC |
| 11. | DXXINA | Intertrochanter Area | DXXINA | Intertrochanter Area |
| 12. | DXXWDBMD | Wards triangle BMD | DXXWDBMD | Wards triangle BMD |
| 13. | DXXWDBMC | Wards triangle BMC | DXXWDBMC | Wards triangle BMC |
| 14. | DXXWDA | Wards triangle Area | DXXWDA | Wards triangle Area |
| 15. | DXAFMRD0 | Calculated value of Vitamin D | DXAFMRD0 | Calculated D Vitamin for femur |
| 16. | OSQ040AA | Under/over 50 when fracd. hip 1st time | OSQ040AA | Under/over 50 when fracd. hip 1st time |
| 17. | OSQ100A | Where fracture occurred | OSQ100A | Where fracture occurred |
| 18. | OSQ160A | Mother had osteoporosis | OSQ160A | Mother had osteoporosis |
| 19. | OSQ160B | Father had osteoporosis | OSQ160B | Father had osteoporosis |
| 20. | OSQ130 | Ever taken prednisone or cortisone daily | OSQ130 | Ever taken prednisone or cortisone daily |
| 21. | OSQ140U | Duration of using prednisone or cortisone? | OSQ140U | Duration of using prednisone or cortisone? |
| 22. | DXXOFBMD | Total femur BMD | OSD030CC | Age when fractured spine 3rd time |
| 23. | DXAFMRK | Calculated value of Vitamin K | OSD030BB | Age when fractured wrist 2nd time |
| 24. | OSQ010A | Broken or fractured a hip | OSD110B | Age when fracture occurred? |
| 25. | OSQ020C | Times broken/fractured spine | OSQ100E | Where fracture occurred |
| 26. | OSQ040BA | Under/over 50 when fracd. wrist 1st time | OSQ150 | Parents ever told had osteoporosis? |
| 27. | OSQ040BB | Second-time wrist fracture younger/older than 50? | OSQ170 | Did mother ever fracture hip? |
| 28. | OSQ040CA | Under/over 50 when fracd. spine 1st time | OSQ180 | Mother's age when fractured hip? |
| 29. | OSD110A | Age when fracture occurred? | OSQ190 | Under/over 50 years old |
| 30. | OSQ120A | Any other fractures? | Osteo_class | Does the Patient ever told you had Osteoporosis |

the most important selected feature using the MI algorithm was if the fracture resulted from severe trauma, and when using RFE, the highest importance registered feature was if the patient ever took prednisone or cortisone daily, as shown in Fig. 3.

## Proposed deep learning models

There were many attempts to create the best models that would better understand the data and discover new patterns and relationships between these features. The most crucial stage in building classification models is the feature selection procedure. After a considerable number of attempts to create the best models to correctly analyze the datasets, only six models were designed using two feature selection methods. It is important to state that this article tested the same architecture of the three models—Sequential DNN, CNN, and

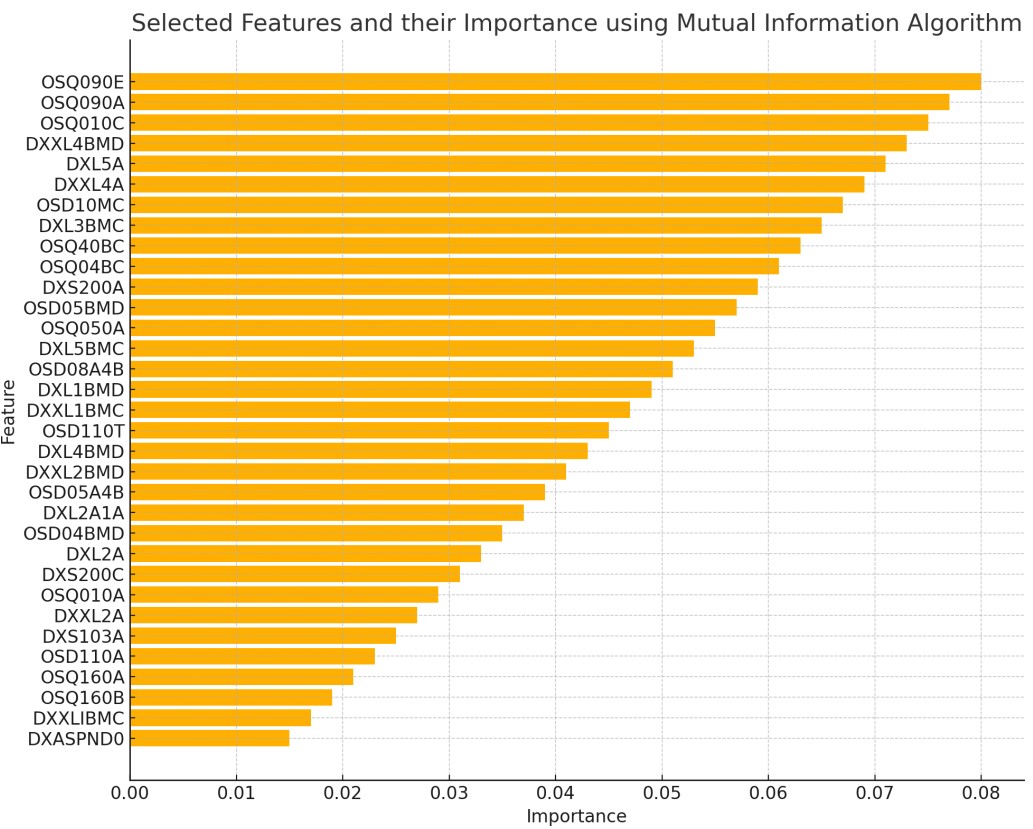

**Figure 2  Selected features using MI algorithm.**

RNN—on both datasets, and it is proven that these models performed at their best for both datasets.

   Each of these models was evaluated using confusion matrix, precision, recall, F1-score, and accuracy after training the models using the SpineOsteo and FemurOsteo datasets for both feature selection algorithms. These calculation metrics formulas are presented as follows:

1. Precision calculates the number of correctly classified patients that have osteoporosis out of all trained values.
   $Precision = \frac{TP}{TP+FP}$ (*Japkowicz, 2006*).

2. Recall measures the number of patients who had osteoporosis that had identified correctly.
   $Recall = \frac{TP}{TP+FN}$ (*George, 2012*).

3. F1-score is the mean value between precision and recall.
   $F1\ score = \frac{2 \cdot TP}{2 \cdot TP+FP+FN} = 2 \cdot \frac{precision \cdot recall}{precision+recall}$ (*Fioravanti et al., 2018*).

4. Accuracy is the proportion of correct predictions (including true positives and true negatives) to total forecasts.
   $Accuracy = \frac{TP+TN}{TP+TN+FP+FN}$ (*Chicco & Rovelli, 2019*).

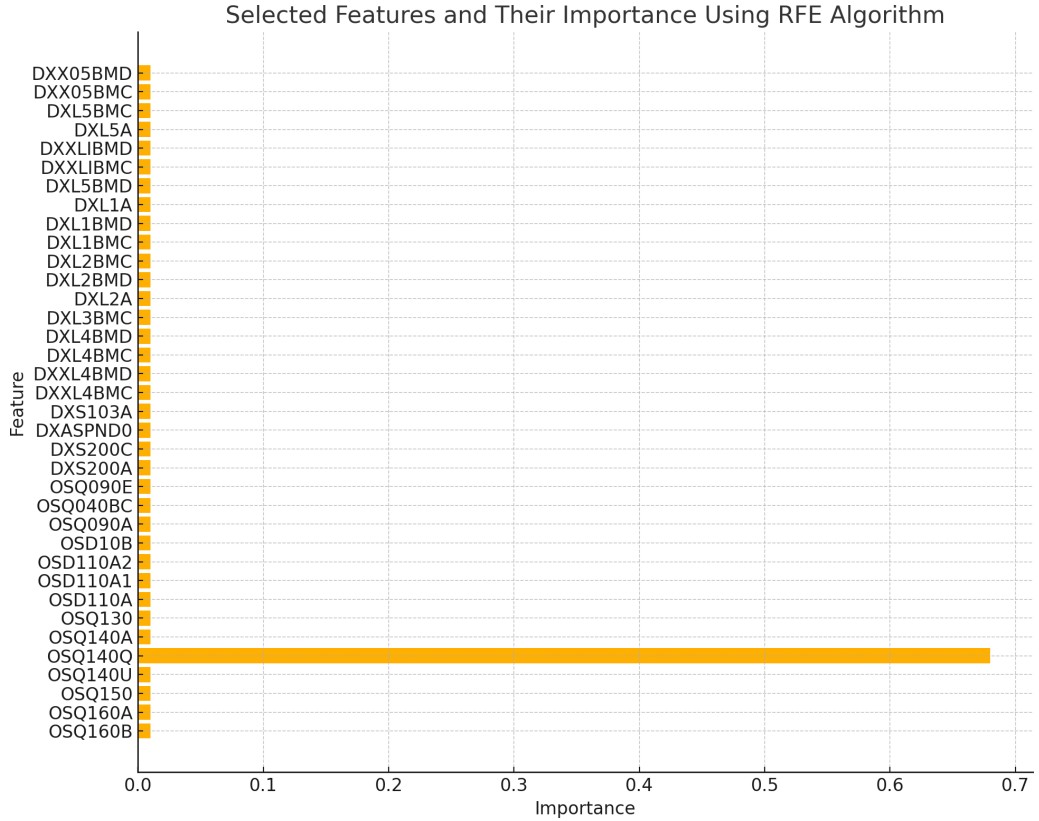

**Figure 3** Selected features using RFE algorithm.

### Sequential DNN model

Machine learning techniques, especially deep neural networks, are a very strong and popular model that has been used in predictive problems, exploring, and explaining the different and complex relationships between variables in the datasets (*Japkowicz, 2006*).

*DNN model architecture.* Layers are added to the model that are fully connected. The first layer consists of 64 units/neurons with rectified linear units (ReLU) as an activation function for the hidden layers, as it has solid biological and mathematical foundations (*Agarap, 2018*). The input dimensions are equal to the number of columns in the selected training data. The second layer has 32 units/neurons that are also activated by ReLU. The third and final layer has two units/neurons with SoftMax activation for the output or the classification layer, as it produces a vector of values that add up to 1.0 and may be used to calculate the target variable's probability (*Brownlee, 2023*) making it appropriate for multi-class classification issues. The model is trained using the training data for a specified number of epochs (30 in this case) and a batch size of 75. It also uses the validation data to monitor the validation accuracy during training before computing the model's loss and accuracy on the test data.

As shown in Table 4 of the Sequential DNN model architecture, the total number of trainable parameters, which are the weights and biases assigned to the model's layers, is

**Table 4  Sequential DNN model architecture.**

| Layer | Type | Output Shape | No. Param in SpineOsteo dataset | No. Param in FemurOsteo dataset |
|---|---|---|---|---|
| Layer 1 | ReLU Dense (fully connected) | 64 neurons | 6,528 | 7,232 |
| Layer 2 | ReLU Dense (fully connected) | 32 neurons | 2,080 | 2,080 |
| Layer 3 | SoftMax Dense (fully connected) | 2 neurons | 66 | 66 |

**Table 5  DNN calculation metrics for SpineOsteo and FemurOsteo datasets with RFE algorithm.**

| Predicted value | SpineOsteo dataset (8,674 trainable parameters) | | | FemurOsteo dataset (9,378 trainable parameters) | | |
|---|---|---|---|---|---|---|
| | Precision | Recall | F1-score | Precision | Recall | F1-score |
| 0 | 99.82% | 98.93% | 99.37% | 99.89% | 99.38% | 99.63% |
| 1 | 92.41% | 98.65% | 95.43% | 94.87% | 99.11% | 96.95% |
| Accuracy | 98.92% | | | 99.47% | | |

**Table 6  DNN confusion metrics for SpineOsteo and FemurOsteo datasets with RFE algorithm.**

| | Predicted 0 | Predicted 1 | Predicted 0 | Predicted 1 |
|---|---|---|---|---|
| Actual 0 | TN = 557 | FP = 1 | TN = 948 | FP = 1 |
| Actual 1 | FN = 6 | TP = 73 | FN = 6 | TP = 111 |

8,674 in the SpineOsteo dataset and 9,378 in the FemurOsteo dataset. This indicates that all parameters were updated through model training.

*Calculation metrics of DNN model.* After creating this model, it is important to check the accuracy of the model and how it fits the two datasets used in this article. The calculation of the confusion matrix, precision, recall, F1-score, and accuracy are used to analyze the DNN Model after training on the SpineOsteo and FemurOsteo datasets. Calculation metrics and confusion metrics for SpineOsteo and FemurOsteo Datasets with RFE Algorithm are presented in Tables 5 and 6, respectively. The predicted values are 0 (class of patients without Osteoporosis) or 1 (class of patients with Osteoporosis).

The same model architecture was used with the same datasets, but a different feature selection algorithm was employed, which is mutual information (MI), as it's powerful in determining the importance of each input variable (*Yang & Moody, 1999*). The Confusion Matrix, accuracy, precision, recall, and F1-score were used as evaluation metrics for the Sequential DNN model. The results are shown in Tables 7 and 8. The predicted values are 0 (class of patients without osteoporosis) or 1 (class of patients with osteoporosis).

### CNN model

CNNs are a type of network that integrates convolution calculations into at least one of its layers. They have become significant for their success in computer vision tasks (*Taner, Oztekin & Duran, 2021*). Training a CNN model poses two challenges: the amount of training data needed, and the time required for network training. This implies that, for a

**Table 7   DNN calculation metrics for SpineOsteo and FemurOsteo datasets with MI algorithm.**

| Predicted value | SpineOsteo Datasets | | | FemurOsteo Dataset | | |
|---|---|---|---|---|---|---|
| | Precision | Recall | F1-score | Precision | Recall | F1-score |
| 0 | 99.82% | 98.93% | 99.37% | 100% | 99.79% | 99.89% |
| 1 | 92.41% | 98.65% | 95.43% | 98.29% | 100% | 99.14% |
| Accuracy | 98.52% | | | 99.89% | | |

**Table 8   DNN confusion metrics for SpineOsteo and FemurOsteo datasets with MI algorithm.**

| | SpineOsteo DNN model | | FemurOsteo DNN model | |
|---|---|---|---|---|
| | Predicted 0 | Predicted 1 | Predicted 0 | Predicted 1 |
| Actual 0 | TN = 557 | FP = 1 | TN = 949 | FP = 0 |
| Actual 1 | FN = 6 | TP = 73 | FN = 2 | TP = 115 |

CNN model to perform well, the training dataset and duration must be sufficient for the CNN to master its tasks effectively (*Abdulnabi et al., 2015*).

*CNN architecture.* The preparation of a CNN model involves training the two datasets, evaluating its performance, and computing the point significance based on the weights of the correlations between the features and the target variable.

A 1D convolutional layer is used in the model. It has 64 filters and a kernel size of three, uses the ReLU activation function, and then adds a Max Pooling layer to reduce the spatial size of the previous layer's output. The output of the preceding layer is flattened into a 1D vector to connect with the fully connected layers when applicable. This Flatten layer is frequently employed in a CNN to restructure the subsequent last layer into a dense layer (*Wang et al., 2020*). Then, another two dense layers (fully connected layers) are added: the first layer has 32 ReLU-activated units, and the second layer contains two SoftMax-activated units.

The CNN model is trained using the training data for 30 epochs and a batch size of 50. It also uses validation data to check validation accuracy throughout training and then computes the CNN model's loss and accuracy on the test data. Table 9 shows the CNN model architecture.

The total number of trainable parameters, which are the weights assigned to the model's layers, is 100,706 in the SpineOsteo dataset and 112,994 in the FemurOsteo dataset. This indicates that all parameters were updated through model training.

*Calculation metrics of CNN model.* For the Spine and Femur datasets, the results of the CNN model with the two selection features algorithms are shown in Tables 10 and 11 for using the RFE algorithm, and Tables 12 and 13 when using the MI algorithm as explained as follows:

### RNN model
A recurrent neural network is a kind of network where the connections between the computing units create a loop allowing it to manage various input sequences by using its

**Table 9  CNN model architecture.**

| Layer # | Type | Output shape SpineOsteo model | Output shape FemurOsteo model | No. Param in SpineOsteo model | No. Param in FemurOsteo model |
|---|---|---|---|---|---|
| Layer 1 | 1D convolutional layer | 99 timesteps and 64 feature or channel | 110 timesteps and 64 feature or channel | 256 | 256 |
| Layer 2 | MaxPooling1D layer | 49 timesteps and 64 feature or channel | 55 timesteps and 64 feature or channel | No additional parameters | No additional parameters |
| Layer 3 | Flatten layer | 49 timesteps and 3,136 feature or channel | 55 timesteps and 3,520 feature or channel | No additional parameters | No additional parameters |
| Layer 4 | ReLU Dense (fully connected) | 3,136 timesteps and 32 feature or channel | 3,520 timesteps and 32 feature or channel | 100,384 | 112,672 |
| Layer 5 | SoftMax Dense (fully connected) | 32 timesteps and 2 feature or channel | 32 timesteps and 2 feature or channel | 66 | 66 |

**Table 10  CNN calculation metrics for SpineOsteo and FemurOsteo Datasets with RFE algorithm.**

| SpineOsteo CNN Model (100,706 trainable parameters) | | | | FemurOsteo CNN Model (112,994 trainable parameters) | | | |
|---|---|---|---|---|---|---|---|
| Predicted value | Precision | Recall | F1-score | Predicted value | Precision | Recall | F1-score |
| 0 | 99.82% | 99.29% | 99.55% | 0 | 99.89% | 100% | 99.94% |
| 1 | 94.49% | 98.68% | 96.77% | 1 | 100% | 99.15% | 99.57% |
| Accuracy | 99.29% | | | Accuracy | 99.94% | | |

**Table 11  CNN confusion metrics for SpineOsteo and FemurOsteo datasets with RFE algorithm.**

| | SpineOsteo CNN model | | FemurOsteo CNN model | |
|---|---|---|---|---|
| | Predicted 0 | Predicted 1 | Predicted 0 | Predicted 1 |
| Actual 0 | TN = 557 | FP = 1 | TN = 948 | FP = 1 |
| Actual 1 | FN = 4 | TP = 75 | FN = 0 | TP = 117 |

internal memory. Each computational unit of an RNN has a changing numbered activation over time along, with weights (*Selvin et al., 2017*).

In this article, an attempt to train an RNN model using LSTM units on the handed data is explained step by step including processes to evaluate its performance and computes the point significance grounded on the weights of the LSTM subcaste. Also, it calculates the correlations between the features and the target variable.

*RNN model architecture.*  A LSTM layer is included in the model as the first layer where its complexity and representational capacity are controlled by 64 units or memory cells. The LSTM layer works well with sequence data and is commonly employed in RNN models. Following that, the second layer in the model is a fully linked layer and it is made up of 32 ReLU-activated units. The last layer is also a fully connected layer with two units with SoftMax activation that is added for multi-class classification. Table 14 illustrates RNN Model Architecture.

**Table 12  CNN calculation metrics for SpineOsteo and FemurOsteo datasets with MI algorithm.**

| | SpineOsteo CNN model | | | | FemurOsteo CNN model | | |
| --- | --- | --- | --- | --- | --- | --- | --- |
| Predicted value | Precision | Recall | F1-score | Predicted value | Precision | Recall | F1-score |
| 0 | 99.64% | 99.64% | 99.64% | 0 | 100% | 99.07% | 99.53% |
| 1 | 97.47% | 97.47% | 97.47% | 1 | 92.31% | 100% | 96.04% |
| Accuracy | 99.15% | | | Accuracy | 99.21% | | |

**Table 13  CNN confusion metrics for SpineOsteo and FemurOsteo datasets with MI algorithm.**

| | SpineOsteo CNN model | | FemurOsteo CNN model | |
| --- | --- | --- | --- | --- |
| | Predicted 0 | Predicted 1 | Predicted 0 | Predicted 1 |
| Actual 0 | TN = 556 | FP = 2 | TN = 949 | FP = 0 |
| Actual 1 | FN = 2 | TP = 77 | FN = 9 | TP = 108 |

**Table 14  RNN model architecture.**

| Layer | Type | Output shape | No. Param in SpineOsteo dataset | No. Param in FemurOsteo dataset |
| --- | --- | --- | --- | --- |
| Layer 1 | LSTM layer | 64 neurons | 42,496 | 45,312 |
| Layer 2 | ReLU Dense (fully connected) | 32 neurons | 2,080 | 2,080 |
| Layer 3 | SoftMax Dense (fully connected) | 2 neurons | 66 | 66 |
| Total number of trainable parameters | | | 44,642 | 47,458 |

*Calculation metrics of RNN model.* The calculation metrics for precision, recall, and F1-score, accuracy, and confusion matrix for RNN model would be illustrated as follow in Tables 15 and 16 for using the RFE algorithm, and Tables 17 and 18 when using the MI algorithm.

Overall, the models perform well with high accuracy, precision, recall, and F1-score. The CNN and RNN models showed the highest precision and recall values, indicating strong performance on the given classification task for both datasets. The learning rate was 0.001 where the models achieved high accuracy which indicates that these models were effectively trained for the datasets as the three models Sequential DNN, CNN, and RNN were designed in the same architecture with two feature selection algorithms.

## DISCUSSION

### Proposed model results

The feature selection algorithms were able to extract the importance and contribution of the features in the classification process. The feature selection process showed that the area measurement of L1 and L3 vertebrae, BMC of L3, L4, and Femur bone, and BMD value of L1 had the highest correlation with the target variable, which was whether the patients had osteoporosis or not. In addition to these values, other medical history information had a significant impact on the classification process, for example, if the mother ever had

**Table 15** RNN calculation metrics for SpineOsteo and FemurOsteo datasets with RFE algorithm.

| | SpineOsteo RNN model | | | | FemurOsteo RNN model | | |
|---|---|---|---|---|---|---|---|
| Predicted value | Precision | Recall | F1-score | Predicted value | Precision | Recall | F1-score |
| 0 | 99.64% | 98.03% | 98.83% | 0 | 99.58% | 99.27% | 99.42% |
| 1 | 86.08% | 97.14% | 91.28% | 1 | 94.02% | 96.59% | 95.29% |
| Accuracy | 97.67% | | | Accuracy | 98.81% | | |

**Table 16** RNN confusion metrics for SpineOsteo and FemurOsteo datasets with RFE algorithm.

| | SpineOsteo RNN model | | FemurOsteo RNN model | |
|---|---|---|---|---|
| | Predicted 0 | Predicted 1 | Predicted 0 | Predicted 1 |
| Actual 0 | TN = 556 | FP = 2 | TN = 945 | FP = 4 |
| Actual 1 | FN = 11 | TP = 68 | FN = 7 | TP = 110 |

**Table 17** RNN calculation metrics for SpineOsteo and FemurOsteo datasets with MI algorithm.

| | SpineOsteo RNN model | | | | FemurOsteo RNN model | | |
|---|---|---|---|---|---|---|---|
| Predicted value | Precision | Recall | F1-score | Predicted value | Precision | Recall | F1-score |
| 0 | 100% | 98.94% | 99.47% | 0 | 100% | 99.68% | 99.84% |
| 1 | 92.41% | 100% | 96.02% | 1 | 97.44% | 100% | 98.71% |
| Accuracy | 99.29% | | | Accuracy | 99.76% | | |

a broken hip, which was the highest value among other medical information, the patient's age when the spine broke for the first time and whether he/she was under or above 50 years old, the age at which the patient's father broke his hip, and if the parents ever had osteoporosis. The two most important features were the age at which the patient broke his/her wrist for the second time, with 27% importance, and the area measurement of the L1 vertebra with 23% importance. Table 19 shows the comparison between the six models where the predicted values are 0 (class of patients without osteoporosis) and 1 (class of patients with osteoporosis).

The models and feature selection methods with the best performance, according to F1-score, as it provides a balanced measurement between recall and precision, are summarized as follows:

\* The best model for the SpineOsteo dataset is the CNN model with the MI feature selection algorithm, while the best model for the FemurOsteo dataset is the CNN model with the RFE feature selection. This concludes that the CNN model is the best performing model regardless of the dataset.

\*\* In terms of feature selection algorithms, it appears that Mutual Information (MI) outperforms Recursive Feature Elimination in the majority of scenarios, as it consistently produces higher F1-scores. As a result, for these models and datasets, MI is the best feature selection technique.

**Table 18  RNN confusion metrics for SpineOsteo and FemurOsteo datasets with MI algorithm.**

| | SpineOsteo RNN model | | FemurOsteo RNN model | |
|---|---|---|---|---|
| | Predicted 0 | Predicted 1 | Predicted 0 | Predicted 1 |
| Actual 0 | TN = 558 | FP = 0 | TN = 949 | FP = 0 |
| Actual 1 | FN = 6 | TP = 73 | FN = 3 | TP = 114 |

**Table 19  Comparison between the six models.**

| Model | Dataset | Feature selection algorithm | Predicted value | Precision | Recall | F1-score | Accuracy |
|---|---|---|---|---|---|---|---|
| Sequential DNN | SpineOsteo | RFE | 0 | 99.82% | 98.93% | 99.37% | 98.92% |
| | | | 1 | 92.41% | 98.65% | 95.43% | |
| | | MI | 0 | 99.82% | 98.93% | 99.37% | 98.52% |
| | | | 1 | 92.41% | 98.65% | 95.43% | |
| | FemurOsteo | RFE | 0 | 99.89% | 99.38% | 99.63% | 99.47% |
| | | | 1 | 94.87% | 99.11% | 96.95% | |
| | | MI | 0 | 100% | 99.79% | 99.89% | 99.89% |
| | | | 1 | 98.29% | 100% | 99.14% | |
| CNN* | SpineOsteo | RFE | 0 | 99.82% | 99.29% | 99.55% | 99.29% |
| | | | 1 | 94.49% | 98.68% | 96.77% | |
| | | MI** | 0 | 99.64% | 99.64% | 99.64% | 99.15% |
| | | | 1 | 97.47% | 97.47% | 97.47% | |
| | FemurOsteo | RFE** | 0 | 99.89% | 100% | 99.94% | 99.94% |
| | | | 1 | 100% | 99.15% | 99.57% | |
| | | MI | 0 | 100% | 99.07% | 99.53% | 99.21% |
| | | | 1 | 92.31% | 100% | 96.04% | |
| RNN | SpineOsteo | RFE | 0 | 99.64% | 98.03% | 98.83% | 97.67% |
| | | | 1 | 86.08% | 97.14% | 91.28% | |
| | | MI | 0 | 100% | 98.94% | 99.47% | 99.29% |
| | | | 1 | 92.41% | 100% | 96.02% | |
| | FemurOsteo | RFE | 0 | 99.58% | 99.27% | 99.42% | 98.81% |
| | | | 1 | 94.02% | 96.59% | 95.29% | |
| | | MI | 0 | 100% | 99.68% | 99.84% | 99.76% |
| | | | 1 | 97.44% | 100% | 98.71% | |

**Notes.**
*The best model for the SpineOsteo dataset is the CNN model with the MI feature selection algorithm, while the best model for the FemurOsteo dataset is the CNN model with the RFE feature selection. This concludes that the CNN model is the best performing model regardless of the dataset.
**In terms of feature selection algorithms, it appears that Mutual Information (MI) outperforms Recursive Feature Elimination in the majority of scenarios, as it consistently produces higher F1 scores. As a result, for these models and datasets, MI is the best feature selection technique

## Comparison

In comparison to other related work, *Alalhareth & Hong (2023)* proposed an intrusion detection system to identify attacks against the Internet of Medical Things (IoMT). They used five classifier models: logistic regression (LR), support vector machines (SVM),

**Table 20** Comparison of our work to the related work presented by *Brownlee (2023)* using MI algorithm.

| Model proposed by M. Alalhareth and S. C. Hong | Size of feature set | | | | | |
|---|---|---|---|---|---|---|
| | 5 | 10 | 20 | 30 | 40 | 45 |
| SVM | 88.4% | 92% | 92.5% | 93.1% | 93% | 93.1% |
| LR | 88.1% | 93.4% | 93.4% | 93.4% | 93.4% | 93.4% |
| RF | 88.4% | 92.5% | 92.4% | 92.4% | 92.5% | 92.5% |
| DT | 85.9% | 94.3% | 94.7% | 94.9% | 94.7% | 94.9% |
| LSTM | 88.4% | 93.4% | 93.4% | 93.3% | 93.4% | 93.4% |
| Proposed models using MI with Feature set of size 30 features | | | | | | |
| | SpineOsteo model | | | FemurOsteo model | | |
| Sequential DNN | 98.52% | | | 99.89% | | |
| CNN | 99.15% | | | 99.21% | | |
| RNN | 99.29% | | | 99.76% | | |

decision tree (DT), random forest (RF), and LSTM, with mutual information feature selection algorithms. The researchers analyzed and evaluated their models based on accuracy, precision, recall, and other metrics. Their approach was to increase the size of the feature set to identify the accuracy of their feature selection algorithm and how it affected their model's accuracy by gradually increasing the number of features from 5 to 45 and noticing that the accuracy of all models increased respectively.

In our models, the number of features was set to 30, as increasing the size of the feature set produced overfitting of the models. The accuracy of the six models was at its best when using the Mutual Information selection algorithm, as shown in Table 19, where it can be noticed that models using the Mutual Information algorithm had higher accuracy and F1-scores than models using the RFE algorithm. Table 20 illustrates the comparison between the models presented in *Alalhareth & Hong (2023)* and our proposed model when using the MI algorithm.

As can be noticed from the comparison above, the accuracy of the proposed models was higher than the other models presented by *Alalhareth & Hong (2023)*, which implies that the type of deep learning models and type of dataset affect the accuracy of models, since some feature types have a great impact and correlation with the target variable. The key point is that the Mutual Information feature selection may minimize the dimensionality of the input data while maintaining the essential information by selecting the most relevant and informative features, thus enhancing the performance and efficiency of our model.

## Authors' novelty

The authors' novelty can be summarized as follows:

1. Non-image data: Most studies into osteoporosis prediction rely on image data, making non-image data the alternative one in most studies. This new approach significantly extends the range of data types usually used for predictive modeling in medical science.

2. First attempt: This is the first attempt at predicting osteoporosis utilizing a combination of medical records and patient questionnaire data. This new combination of innovative data sources is a new perspective on osteoporosis prediction.

3. Deep learning models: This study discovers how to use advanced deep learning models, namely, sequential deep neural networks, convolutional neural networks, and recurrent neural networks for osteoporosis prediction. This has been done novelly and opens the application of state-of-the-art deep learning techniques on non-image medical data.

4. Effect of feature selection: The study demonstrates how feature selection can help create a better accuracy model. A comparison of MI *versus* RFE methods used for feature selection is a way of finding possibilities for improvement when developing the predictive model in the health domain.

5. Family medical history: This study highlights the significant role of family medical history in Osteoporosis prediction. This remark points to the potential benefits of integrating detailed medical history information into predictive models in achieving improvements in accuracy and relevance.

These contributions exemplify how deep learning applied to medical data analysis is novel, leading to new research lines and clinical practice for osteoporosis prediction.

## CONCLUSION

According to our findings, Sequential DNNs, CNNs, and RNNs are effective in classifying non-image medical data. These findings indicate that the models have been trained to categorize data with high accuracy and low error, as the accuracy progressively grows with each cycle of the model's training, indicating that the models are improving at accurately identifying the data.

This research showcases the advancements and novel ideas introduced by the authors in the field of medical data analysis by utilizing deep learning methods on non-image medical data and has ventured into a fresh approach that broadens the scope of medical data analysis. The study examined data obtained from NHANES ensuring a comprehensive dataset for efficient model training and validation, which highlights the impact of learning techniques in enhancing medical diagnostic procedures. The preparation and evaluation of the six models using both datasets highlighted the significance of tailoring the datasets according to the classification task at hand. To our best knowledge, this study marks the first attempt to forecast osteoporosis utilizing medical records and patient survey details.

Incorporating MI and RFE for feature selection has shown enhancements in model performance, emphasizing feature selection's role in creating precise predictive models. The outcomes of this study hold implications for healthcare particularly concerning osteoporosis prediction. By improving the precision and dependability of models, this study can assist healthcare professionals in detection and treatment planning ultimately leading to improved patient outcomes.

Deep learning played an important role in this research, showing the importance of family medical history and one's health concerning osteoporosis. Important features significantly correlated with osteoporosis, such as bone measurements (*e.g.*, area

measurements of vertebrae and bone mineral content), and medical history information (*e.g.*, history of fractures, family history of osteoporosis, and prior use of medications like prednisone or cortisone). The Sequential DNN, CNN, and RNN models all used a set of selected features that included whether the patient's father had osteoporosis, whether the mother ever broke a hip when she was older or younger than 50, and whether a patient ever took vitamin K or D. The research underscored the significant role of family medical history and individual health records in predicting osteoporosis. The findings pave the way for more precise diagnostic tools and a better understanding of the relation between medical history and osteoporosis risk. The study was limited to data from NHANES 2017–2020, which may not capture all possible variations in patient demographics and medical histories. The study does not address the integration of these models into real-time clinical decision support systems, leaving a gap in understanding how they can be implemented in actual clinical workflows and practice.

Future work could include more comprehensive datasets to improve model robustness. Also, future work would be integrating these models into clinical decision support systems (CDSS) could assist healthcare providers in making more informed decisions by providing real-time risk assessments and predictions. Another future contribution can be implementing real-time data processing capabilities that could allow for the continuous update and improvement of the models based on newly available patient data.

### Funding
The authors received no funding for this work.

### Competing Interests
The authors declare there are no competing interests.

### Author Contributions
- Zahraa Noor Aldeen M. Shams Alden conceived and designed the experiments, performed the experiments, analyzed the data, performed the computation work, prepared figures and/or tables, authored or reviewed drafts of the article, and approved the final draft.
- Oguz Ata analyzed the data, authored or reviewed drafts of the article, and approved the final draft.

### Data Availability
The code is available in the Supplemental File.

The Femur, Spine and Questionnaire datasets were taken from the 2017–March 2020 Pre-Pandemic Questionnaire Data which is available from the National Center for Health Statistics: https://wwwn.cdc.gov/nchs/nhanes/search/datapage.aspx?Component=Questionnaire&Cycle=2017-2020.

1. The Femur and Spine datasets under Examination Data group where you can find Dual-Energy X-ray Absorptiometry - Femur and Dual-Energy X-ray Absorptiometry - Spine datasets information.

2. The Questionnaire - Osteoporosis datasets under Questionnaire Data group.

## Supplemental Information

Supplemental information for this article can be found online at http://dx.doi.org/10.7717/peerj-cs.2338#supplemental-information.

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
