# Peer review of "A comprehensive analysis and performance evaluation for osteoporosis prediction models"

_PeerJ Computer Science, doi:10.7717/peerj-cs.2338_

## Round 0.1 · original submission · Major Revisions

After carefully considering the reviews and assessing your manuscript, I am pleased to inform you that we would like to invite you to revise and resubmit your manuscript for further consideration. The reviewers have provided constructive comments that will help strengthen your work. Please address each of these points thoroughly in your revised manuscript. Additionally, ensure that you provide a detailed response letter outlining how you have addressed each comment raised by the reviewers. This will help the reviewers and myself to evaluate the changes made to the manuscript.

Some of the my comments are:

1) Title is too long that may be shortened.
2) English language needs to be improved.

Reviewer 1 ·

Basic reporting

- This manuscript describes about the process of analyzing the medical data set and creating
50 classification models using Sequential DNN, CNN, and RNN models.

- Introduction section needs to be more in detail. It is even shorter than Abstract in the present form.

- Literature review section needs to be modified. Related research findings should be summarized in a table properly in year-wise order.

- Figures 1 to 3 seem like a screenshot. Needs improvement.

- Overall, the structure of the paper is not well-organised.

Experimental design

- Experimental section is missing.

- All parameter need to be describe in tables in the experimental section.

- 5.1.2 Calculation Metrics should be in experimental section.

- Details discussion should be in the experimental section.

Validity of the findings

Novelty should be highlighted in the literature review section.

Also, highlight current status and limitations.

- Future scope should be in included in the Conclusion section.

Additional comments

Major revision is required to improve the present form of the manuscript.

Reviewer 2 ·

Basic reporting

The study represents an endeavour and first attempt at predicting osteoporosis based on a blend of data and patient questionnaire responses. The six developed models showcased accuracy levels coupled with error rates. However, the explanation of the contribution in the paper is not adequate. The whole chapter needs significant improvement. The content of the chapter is ambiguous. The introduction is not well written, and the authors are unable to justify their contributions. Moreover, the results of the proposed work are not clear. The title of the paper is too long.

Experimental design

The authors should clarify how they have collected the data. It is very small which is not suitable for deep learning models.
Questionnaires may introduce biases in the experiment, which is not good for such types of studies.

Validity of the findings

Authors are unable to claim their contributions in the related works section.

Additional comments

1. Figures related to models and frameworks are missing and not cited in the text.
2. Authors did not cite work properly as it is missing in many places.
3. Most of the content is copied and edited, which could not be acceptable.
4. Authors should clarify how they have collected the data of patients. It is very small which is not suitable for deep learning models.
5. The authors did not provide future directions, which would guide and motivate other researchers to do work in the said domain.
6. Authors are advised to provide a details discussion of the models.

---

## Round 0.2 · Major Revisions

Reviewers are not satisfied with the revision made. You are required to go through each and every comment, and address them properly.

Reviewer 1 ·

Basic reporting

1. Authors have addressed comments in the revised manuscript, but still minor revision is required.

2. Expand related works in more detail. Add a sub-section 2.1 as, "Current status and limitation" and highlight your novelty the related work section-2.

Experimental design

I am still not satisfied with the response of the comments for experimental section. It is not properly addressed.

1. Calculation metrics are not replaced with evaluation metrics. Why?

2. Why there is two sub-sections for calculation metrics (3.3.2.1.2 Calculation Metrics and .3.2.2.2 CNN Calculation Metrics). It should be in one with Evaluation metrics name, where in all evaluation metrics should be defined.

3. Why methodology and experimental sections are not separated yet. It must be separated.

Validity of the findings

1. Highlight future work as a separate section after conclusion.

2. Check typos and grammatical errors again.

Additional comments

Revision required (Minor)

Reviewer 2 ·

Basic reporting

The revised manuscript fails to meet the requirement of high-quality standards. It is lacking in various aspects such as novelty, experiment, results etc.

Experimental design

The experiment is not up to the mark.

Validity of the findings

Authors are unable to claim their findings in the manuscript.

Additional comments

Most of the content is copied and edited, which could not be acceptable.

---

## Round 0.3 · Major Revisions

There are some changes required:

1) In the title, instead of 'Model", it should be "Models". Because multiple models were compared.
2) The abstract has too many unnecessary statements and missing important statistical results and findings. Pl modify it. Also, it must be written in single paragraph.
3) Related work is very limited. Authors must enhance this section and incorporate recent related papers to show the state of the art in the field. Also, mention sufficient motivation behind such a comparative study.
4) There are lots of linguistic errors, e.g. line 71, "This will ....", line 77, "It will ...", etc. Authors need to thoroughly check English language, and correct it.
5) The methodology section has not been written sufficiently. Authors must present sufficient background of deep learning models considered in this comparative study.
6) Section 5.1 is at inappproriate place. It should go at the end of the Introduction Section.
7) Pl check Table 1. It is not listing features, rather presenting the contributions of selected papers.
8) Some of the refernces have been writtein separately, just after Table 1. What is the reason for such a separate list of references?
9) Only CNN have been applied, what about other deep learning models (e.g. RNN) used for comparison?
10) Manuscript needs major revision, written and presented in a novice way. I suggest to get it reviewed and corrected by a senior researcher from your lab.

Good luck.

Reviewer 1 ·

Basic reporting

Authors have addressed all comments.

Experimental design

Comments are addressed in the revised manuscript.

Validity of the findings

Accepted

Additional comments

Accept

---

## Round 0.4 · Major Revisions

Most of the suggested comments were not addressed in the revision. The authors have neither addressed those comments, nor provided reasonable rebuttal. The revision has been made very casually. in the current form, manuscript can not be accepted for the publication.

---

## Round 0.5 · accepted · Accept

I am pleased to inform you that your paper has been accepted for publication in PeerJ Computer Science. Your manuscript has undergone rigorous peer review, and I am delighted to say that it has been met with praise from our reviewers and editorial team. Your research makes a significant contribution to the field, and we believe it will be of great interest to our readership. On behalf of the editorial board, I extend our warmest congratulations to you.